# Enhancing Computation Efficiency in Large Language Models through Weight and Activation Quantization

**Janghwan Lee**[1][*] **Minsoo Kim**[1][*] **Seungcheol Baek**[2], **Seok Joong Hwang**[2],
**Wonyong Sung**[3] and **Jungwook Choi**[1][†]

{hwanii0288, minsoo2333}@hanyang.ac.kr, {bsc11235, nzthing}@sapeon.com

wysung@snu.ac.kr, choij@hanyang.ac.kr

[1]Hanyang University, [2]SAPEON Korea Inc., [3]Seoul National University
Republic of Korea

## Abstract

Large Language Models (LLMs) are proficient in natural language processing tasks, but their deployment is often restricted by extensive parameter sizes and computational demands. This paper focuses on post-training quantization (PTQ) in LLMs, specifically 4-bit weight and 8-bit activation (W4A8) quantization, to enhance computational efficiency—a topic less explored compared to weight-only quantization. We present two innovative techniques: activation-quantization-aware scaling (AQAS) and sequence-length-aware calibration (SLAC) to enhance PTQ by considering the combined effects on weights and activations and aligning calibration sequence lengths to target tasks. Moreover, we introduce dINT, a hybrid data format combining integer and denormal representations, to address the underflow issue in W4A8 quantization, where small values are rounded to zero. Through rigorous evaluations of LLMs, including OPT and LLaMA, we demonstrate that our techniques significantly boost task accuracies to levels comparable with full-precision models. By developing arithmetic units compatible with dINT, we further confirm that our methods yield a $2\times$ hardware efficiency improvement compared to 8-bit integer MAC unit.

## 1 Introduction

Large language models (LLMs) have achieved breakthroughs in many natural language processing tasks such as translation, summarization, reasoning, and conversation, often matching or exceeding human performance (Zhang et al., 2022; Touvron et al., 2023; Chowdhery et al., 2022; Brown et al., 2020; OpenAI, 2023). However, the extensive parameters of LLMs present deployment challenges due to the high memory bandwidth needed for high throughput inference. Post-training quantization (PTQ) addresses this by "compressing" weight pa-

---

[*]equal contribution   [†]corresponding author

rameters, significantly reducing memory requirements and enhancing GPU performance by alleviating memory bandwidth bottlenecks (Frantar et al., 2023; Lin et al., 2023; Lee et al., 2023a). Nevertheless, LLMs' computational complexity remains a concern. For example, GPT-3 (Brown et al., 2020) requires at least 350 GFLOPs of computation for a single token, but PTQ methods often revert compressed weights to higher precisions like 16-bit floating-point (FP16) for computation, which is inefficient given the resource demands of multiply-accumulate (MAC) operations. With computing platforms evolving through high-bandwidth memory (Gurumurthi et al., 2021) and processing-in-memory (Kim et al., 2021; He et al., 2020) to resolve the memory bandwidth bottleneck, addressing LLMs' computational needs becomes more imperative.

A PTQ strategy that effectively quantizes both weights and activations is thus appealing as it reduces the hardware complexity of MAC units, enhancing computational throughput (Sun et al., 2019; Dettmers et al., 2022; Xiao et al., 2022). PTQ research specific to LLM's computation efficiency is growing, focusing on utilizing INT8-INT8 MAC units, common in GPUs (Andersch et al., 2022). LLM.Int8 (Dettmers et al., 2022), for instance, used INT8 quantization for weights and activations, but directed activation outliers through an FP16 datapath, isolating them. SmoothQuant (Xiao et al., 2022) extended this by employing activation channel scaling to target outliers and adjusting corresponding weights for balanced quantization. However, these studies do not address challenges faced when weights are reduced to 4 bits, revealing an unexplored area for combined effects on weight and activation quantization.

This paper delves into the challenges of post-training quantization (PTQ) for both weights and activations in large language models (LLMs). We pinpoint two primary hurdles in achieving efficient

4-bit weight and 8-bit activation (W4A8) quantization. First, LLMs like OPT (Zhang et al., 2022) and LLaMA (Touvron et al., 2023) have distinct weight and activation range characteristics, making existing PTQ methods unsuitable for universal use. For example, AWQ's (Lin et al., 2023) activation-aware scaling makes activations prone to quantization errors, while OPTQ's (Frantar et al., 2023) weight calibration struggles with varying activation ranges. We propose two novel solutions for this first hurdle: activation-quantization-aware scaling (AQAS) and sequence-length-aware calibration (SLAC). AQAS optimizes quantization scales by jointly considering weights and activations, yielding balanced quantization. SLAC aligns the sequence length of the application task with that of the PTQ calibration dataset, mitigating the impact of variations in activation diversity, which significantly affects the PTQ calibration process.

Second, we observe that *underflow*, where small-magnitude values round to zero, severely impacts W4A8 quantization in LLMs because the quantization error associated with values rounding to zero constitutes a significant portion of the output error. While underflow is a well-known issue in reduced-precision formats for deep neural networks (DNNs) (Sun et al., 2019, 2020; Chmiel et al., 2022; Jin et al., 2022), previous PTQ research in LLMs mainly focuses on outliers, neglecting underflow. We discover that standard INT4 representation discards crucial small-magnitude weights when multiplied with activations. As existing data formats like integer, floating-point, or logarithmic formats are inadequate for this underflow issue, we introduce dINT, a new integer format with denormal representation. dINT merges the uniform coverage of integers with the denormal of floating-points, effectively mitigating underflow and improving accuracy. We also propose a MAC unit supporting dINT to ensure hardware efficiency.

We evaluate AQAS, SLAC, and dINT on OPT and LLaMA, focusing on language modeling, zero-shot reasoning, and 5-shot in-context learning. The results show that integrating these methods for W4A8 PTQ significantly improves task accuracies for both OPT and LLaMA across a diverse set of benchmarks (Wikitext, Common Sense Question Answering (CSQA), and Massive Multitask Language Understanding (MMLU)) and the model sizes ranging from 125M to 65B parameters.

## 2 Background

### 2.1 Weight-only PTQ for LLMs

Various weight-only PTQ techniques have emerged to alleviate memory-bandwidth constraints in LLM inference by compressing weights to 4 bits while maintaining accuracy (Park et al., 2023; Kwon et al., 2022; Frantar et al., 2023; Lin et al., 2023; Lee et al., 2023a). For example, OPTQ (Frantar et al., 2023) reduces output distortion from column-wise weight quantization by sequentially updating unquantized weights using activation Hessians. AWQ (Lin et al., 2023) scales weights according to activation magnitudes for improved quantization, while OWQ (Lee et al., 2023a) and SPQR (Dettmers et al., 2023) isolate sensitive weights, retaining them at higher precision. However, these approaches entail high-precision computations and complex arithmetic units. We demonstrate that these weight compression methods are sub-optimal for activation quantization in common LLMs, often exacerbating challenges by ignoring activation dynamics. Consequently, we introduce advanced techniques specifically designed to address these intricacies, enhancing weight quantization accuracy when the activation is also quantized.

### 2.2 Weight and Activation PTQ for LLMs

Quantizing both weights and activations enables the use of lower-precision MAC units, significantly saving logic area and power consumption (Horowitz, 2014). As such, many studies aim to reduce DNN's computational burden (Sun et al., 2019; Lee et al., 2023b), especially in LLMs (Dettmers et al., 2022; Xiao et al., 2022; Liu et al., 2023; Bondarenko et al., 2021). For instance, LLM.Int8 (Dettmers et al., 2022) and SmoothQuant (Xiao et al., 2022) employ GPU-supported INT8-INT8 MAC operations for efficiency, with LLM.Int8 processing outliers separately and SmoothQuant adjusting activations and weights. Additionally, (Liu et al., 2023; Bondarenko et al., 2021) employ quantization-aware fine-tuning for further reductions to W4A8 or W4A4, but face noticeable accuracy losses despite expensive fine-tuning. This paper proposes novel solutions that address the accuracy drop in combined weight and activation quantization with bit-precision down to W4A8, achieving superior results compared to prior works without fine-tuning.

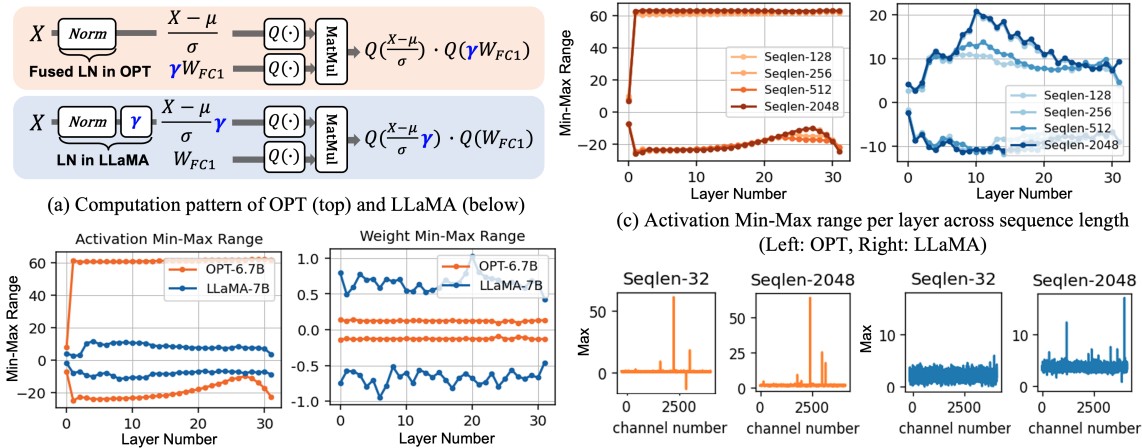

(a) Computation pattern of OPT (top) and LLaMA (below)

(b) Min-Max range of input activation and weight per layer

(c) Activation Min-Max range per layer across sequence length (Left: OPT, Right: LLaMA)

(d) Max values of per-channel input activation (Left: OPT, Right: LLaMA)

Figure 1: (a) Illustration of fused-layernorm (fused-LN) in OPT (top) and layernorm (LN) in LLaMA (bottom) computation patterns within a Transformer layer. Note that two computation patterns yield ths same output if computed in full-precision, but they deviate when activation and weight are quantized. (b) Min-Max range of input activations (left) and weight (right) as operands of matrix multiplication. (c) Min-Max range of input activation varying sequence length from 128 to 2048 (Orange: OPT-6.7B, Blue: LLaMA-7B). (d) Max values of per-channel input activation for OPT-6.7B (left) and LLaMA-7B (right) for different input sequence lengths (32 and 2048).

## 2.3 Underflow for Reduced-Precision LLMs

*Underflow*, the numerical error from small values rounding to zero due to limited bit-precision, has been actively studied as a critical issue in reduced-precision DNN training. For instance, (Sun et al., 2019) counters underflow in 8-bit floating-point by adjusting the exponent bias, (Sun et al., 2020) utilizes a radix-4 format to represent wider magnitude ranges in 4-bit floating-point (FP4), and (Chmiel et al., 2022) uses stochastic underflow to address biased quantization in FP4 gradients. In fixed-point representation, (Jin et al., 2022) explores optimal formats by analyzing underflow and overflow trade-offs based on fractional length. Contrary to these studies focusing on the training phase, our paper investigates underflow's impact on PTQ of LLMs for the first time and introduces an enhanced integer format to combat it.

## 3 Improving PTQ for Weight and Activation Quantization

We aim to advance LLM quantization beyond the realms of 4-bit weight-only PTQ or W8A8 PTQ by investigating the combined effects of weight and activation quantization. When quantizing both weight and activation, it is important to note that LLMs display distinct weight and activation characteristics. For example, OPT has been found to have 0.1% activation outliers by (Dettmers et al., 2022), whereas GLM-130B (Zeng et al., 2023) reported 30% of outliers in its model. In the context of weight, due

to varied weight distributions across models, OPT-66B experiences a substantial perplexity increase in the wikitext benchmark with INT4 weights, soaring from 9.34 to 110 (Frantar et al., 2023), whereas GLM-130B shows no performance degradation on the MMLU benchmark when INT4 weights are applied (Zeng et al., 2023). We posit that these discrepancies arise from variances in pre-training configurations such as datasets, learning rates, layer structures, and self-attention directionality, as well as options designed for efficient inference, such as operation fusion techniques like layernorm fusion. Significantly, existing PTQ research has overlooked these unique traits intrinsic to each model that are pivotal for the combined optimization of activation and weight quantization. Therefore, we delve into the weight and activation distributions of widely-used OPT and LLaMA models during quantization to understand PTQ limitations and develop novel methods to address them.

### 3.1 Model Analysis: OPT vs. LLaMA

To understand the adverse effects of quantization on restricting dynamic range, we examine the minimum and maximum values (Min-Max range) across the layers of LLMs. Fig. 1(a) illustrates the computation patterns within a layer of LLMs and Fig. 1(b) displays Min-Max range of activations (left) and weights (right) as operands of matrix multiplication for each FC layer in OPT and LLaMA. Notably, there are contrasting trends in

Min-Max ranges; OPT has a broad activation range but a narrow weight range, while LLaMA exhibits the opposite. This distinction stems from the way these LLMs process activations at layernorm. As depicted in Fig. 1(a), in OPT, the layernorm parameters are fused to the subsequent FC layer's weights (Fig. 1(a) top), allowing only normalized activation to enter the FC layer. Conversely, layernorm is not fused in LLaMA (Fig. 1(a) below), resulting in scaled activation as input to FC layers. Although layernorm fusion preserves functionality in full-precision computation, this presence or absence of layernorm fusion in activation processing contributes to significantly distinct behaviors under quantization, as will be discussed in the following sections.

Another insightful finding from our model analysis is the variation in activation diversity based on *sequence lengths*. Fig. 1(c) displays the Min-Max range as sequence length varies from 128 to 2048 (Orange: OPT-6.7B, Blue: LLaMA-7B). Notably, OPT's activation range remains stable across sequence lengths, while LLaMA's activation range expands, suggesting challenges in range calibration for quantization. Fig. 1(d) contrasts maximum values per channel for OPT and LLaMA at varying sequence lengths. OPT displays consistent outliers at the same channels, dominating its activation dynamic ranges. In contrast, LLaMA's outliers increase in magnitude and shift across channels, indicating varied activation dynamic ranges. This distinction in activation diversity is significant for quantization. While PTQ generally presumes consistent dynamic ranges for calibrating quantization ranges "offline", these findings emphasize the necessity of considering distinct activation dynamic range and incorporating sequence length into calibration. The following sections discuss methods to optimize weight and activation quantization, building on these model-specific insights.

## 3.2 Activation-Quantization-Aware Scaling

The distinct properties of outliers in weights and activations illustrated in Fig. 1(b) pose challenges of applying prior scaling techniques. Fig. 2 illustrates the absolute maximum of (a) input activations and (b) weights at the "Key" layers (for self-attention) in OPT-6.7B when different scaling methods are applied. Specifically, SmoothQuant (Xiao et al., 2022) (SQ) scales activation for 8-bit quantization, but descales weights, resulting in a more diverse

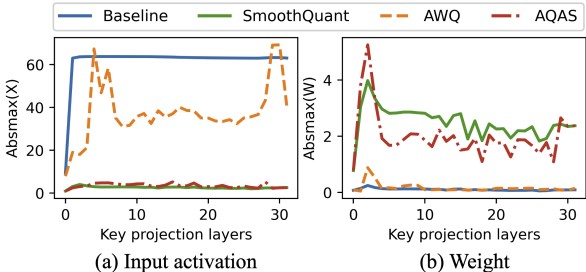

Figure 2: Absolute max value of (a) input activation and (b) weight after scaling by each method (OPT-6.7B). We observed that these trends were significantly pronounced in OPT models due to large outliers. (See Fig. 6 for the same plot for LLaMA.)

and quantization-sensitive range for weight. On the other hand, AWQ (Lin et al., 2023) scales weights for 4-bit quantization but significantly increases activation diversity, making activation quantization problematic. In other words, the existing scaling-based PTQ techniques such as SQ and AWQ cannot resolve the issue of conflicting trends in activation and weight outliers. To address this, we introduce activation-quantization-aware scaling (AQAS), a hybrid of SQ and AWQ. AQAS aims to find scaling values that minimize the output error caused by quantized weights and activations. We use mean squared error (MSE) loss as the objective function, aligning with previous studies on layer-wise optimization (Nagel et al., 2020; Frantar et al., 2022). Our objective function is as follows:

$$\underset{\mathbf{s}}{\mathrm{argmin}} \, ||Q(\mathbf{W} \cdot \mathrm{diag}(\mathbf{s}))Q(\mathrm{diag}(\mathbf{s})^{-1} \cdot \mathbf{X}) - \mathbf{W}\mathbf{X}||_2^2 \tag{1}$$

We define the weight $\mathbf{W} \in \mathbb{R}^{M \times C}$, scale factor $\mathbf{s} \in \mathbb{R}^C$, and activation $\mathbf{X} \in \mathbb{R}^{C \times T}$, where $M$ represents the output feature dimension, $C$ represents the input feature dimension, and $T$ denotes the number of tokens. Fig. 2 demonstrates that AQAS considers activation quantization's impact to adjust activation magnitudes, easing activation quantization. Additionally, as compared to SQ, AQAS adjusts weight magnitudes more moderately, making 4-bit weight quantization feasible.

## 3.3 Sequence-Length-Aware Calibration

As shown in Fig. 1(c), variation in activation diversity depending on the sequence length affects the quantization performance. Specifically, weight-update-based quantization like OPTQ (Frantar et al., 2023) struggles with models like LLaMA that have increasing activation diversity during calibration. To delve deeper into this phenomenon,

| Sequence Length | 64 | 128 | 512 | 2048 | $std$ |
|---|---|---|---|---|---|
| FP (LLaMA-7B) | | 70.92 | | | - |
| INT4 | | 69.37 | | | - |
| INT4 OPTQ | 68.14 | **69.89** | 65.96 | 65.19 | **2.54** |
| INT4 AQAS+OPTQ | 69.99 | **70.48** | 68.92 | 70.20 | 0.68 |

Table 1: W4A8 Quantization zero-shot evaluation of CommonSenseQA (average score of PIQA, Winogrande and Arc_easy, default calibration sequence length is 2048)

we analyze the approach adopted by OPTQ, which employs weight adjustments in response to quantization error using activation Hessian, formulated as follows (Frantar et al., 2023):

$$\delta_F = -\frac{w_q - \text{quant}(w_q)}{[\mathbf{H}_F^{-1}]_{qq}} \cdot (\mathbf{H}_F^{-1})_{:,q} \quad (2)$$

$$\mathbf{H}_i = \frac{\partial^2 E}{\partial \mathbf{W}_{i,:}^2} = 2\mathbf{X}\mathbf{X}^T, \quad (3)$$

where $\mathbf{X}$ denotes the layer input activation, $\mathbf{W}$ is weights of linear layer, $w_q$ is weight element to quantize, and $\delta$ denotes optimal weight update recovering quantization error. We examine the weight update ratio, $(\mathbf{H}_F^{-1})_{:,q}/[\mathbf{H}_F^{-1}]_{qq}$, representing the second derivative of quantization error ($E$), to assess changes in weights due to OPTQ. Fig. 3(a) shows the weight update ratio for OPT and LLaMA with varying calibration sequence lengths. OPT remains relatively consistent, while LLaMA displays varying weight update ratios for varying sequence length, suggesting activation diversity affects OPTQ's weight updates.

This sensitivity of OPTQ updates prompts us to further explore its implications for performance. We evaluate the zero-shot performance of OPTQ for W4A8 quantization by varying the calibration sequence length on PIQA, Winogrande, and Arc_easy tasks from CSQA (Bisk et al., 2019; Sakaguchi et al., 2019; Clark et al., 2018), which have sequence lengths ranging from tens to hundreds (note that the type of calibration dataset was kept consistent). Table 1 reveals that when the calibration sequence length (e.g., 512 or 2048) significantly deviates from task's sequence lengths, OPTQ's performance suffers (up to 4% degradation), even falling below basic nearest-rounding quantization. However, when the sequence lengths are aligned (e.g., 64 or 128), OPTQ performs exceptionally well.

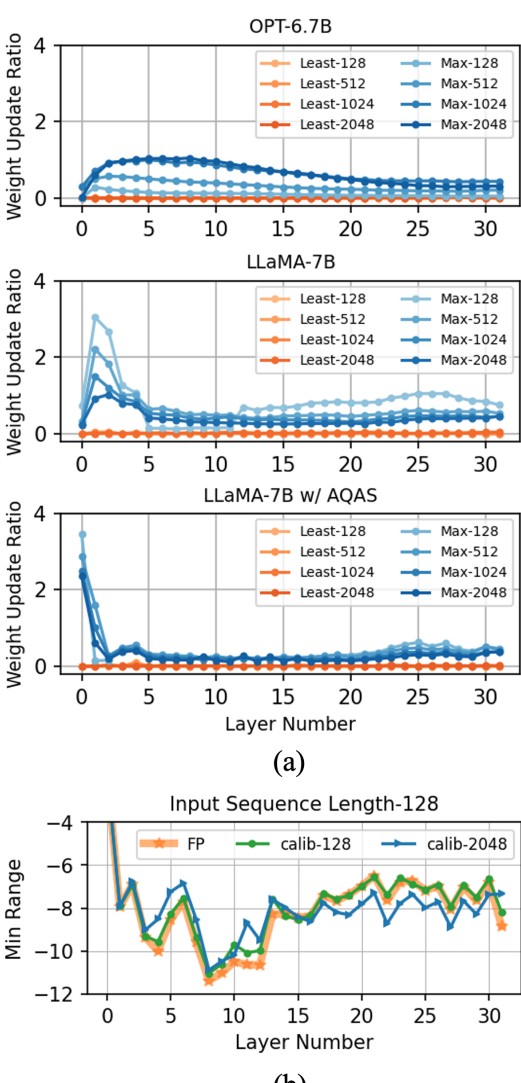

Figure 3: (a) Comparison of weight update ratio in Eq. 2 in OPT-6.7B, LLaMA-7B, and LLaMA-7B with AQAS scaling. (b) Minimum input activation range for the query layer in three models: W4A8 (calibrated with 128 and 2048 sequence lengths) and full-precision (FP), all evaluated under an input sequence length of 128.

The large standard deviation in accuracies for matching versus non-matching sequence lengths suggests that LLaMA's activation diversity substantially impacts OPTQ's accuracy. To mitigate this, we propose the sequence-length-aware calibration (SLAC) method. This approach involves determining the expected sequence length during the target task's inference phase and aligning the sequence length of the calibration dataset accordingly. Such a task-specific PTQ calibration process enhances the robustness and accuracy of the model's inference. The efficacy of SLAC, particularly in the CSQA benchmark, is substantiated by experiments detailed in Sec. 5.3.

The effectiveness of the SLAC method is evi-

dent when comparing the dynamic range of quantized models with their full-precision counterparts. Fig. 3 (b) demonstrates that using calibration data aligned with the input sequence length (calib-128) results in a dynamic range more consistent with that of the full-precision model (FP), unlike models calibrated with mismatched sequence lengths (calib-2048).

Integrating SLAC with AQAS effectively enhances weight and activation quantization. As illustrated in Fig. 3(a), AQAS efficiently mitigates the sensitivity to input sequence length regarding weight updates. Moreover, Table 1 shows that the standard deviation related to the calibration dataset's length is significantly reduced from 2.54 to 0.68 through AQAS. Consequently, combining AQAS with OPTQ proves advantageous for inferences across diverse sequence lengths, and employing the SLAC method for calibration according to the target dataset's sequence length further bolsters performance.

## 4 Overcoming PTQ Underflow for LLMs

By employing AQAS to address activation quantization errors in weight scaling, and utilizing SLAC to align the sequence length of the calibration dataset with that of the target inference, we achieve a substantial improvement in the performance of our W4A8 models. However, we encounter persistent performance degradation issues. In this section, we unveil "underflow" issues as a previously overlooked cause of accuracy degradation in PTQ applied to LLMs and propose a new numerical format to mitigate this problem.

### 4.1 Observations

We identify underflow as a main contributor to performance degradation. To dissect the causes of degradation when converting the weights of the scaled model to 4-bit, we split the quantization error into two parts: *rounding error* ($\Delta_r$) and *underflow error* ($\Delta_u$). The rounding error accounts for the error when the quantized value is non-zero, whereas the underflow error represents the error occurring when the quantized value rounds to zero. By considering the total error ($\Delta$) induced by quantization as a combination of $\Delta_u$ and $\Delta_r$, we can express the expected output quantization error as

follows:

$$\mathbf{E}[(\mathbf{WX} - (\mathbf{W} + \Delta_u + \Delta_r)\mathbf{X})^2] \qquad (4)$$
$$= \mathbf{E}[(\Delta_u\mathbf{X})^2] + \mathbf{E}[(\Delta_r\mathbf{X})^2] + \mathbf{E}[2(\Delta_u\mathbf{X}\Delta_r\mathbf{X})].$$

Fig. 4(a) exemplifies the underflow issues, illustrating the distinct impacts of quantization errors on final model accuracy, measured as perplexity. The figure highlights that setting small values near zero to exactly zero, while leaving other values unquantized, impairs performance. In contrast, quantizing larger values and precisely representing those near zero significantly improve accuracy. Fig. 4(b) provides a breakdown of error terms across layers in OPT W4A8, indicating a correlation between high total error and substantial underflow error. This underlines the necessity for a method that effectively addresses underflow errors.

### 4.2 Integer with Denormal Representation

Inspired by our observations and the denormal numbers in floating-point representation, we introduce a new integer format called integer with denormal representation (dINT). As illustrated by Fig. 4(c), dINT uses two bins around zero to ensure lower magnitudes are effectively represented. In $b$-bit quantization, two values are reserved for special cases, so the quantization range represents integers from 0 to $2^b - 3$. These special values in dINT have magnitudes equal to *half of the chosen step size*, which is a power of two to enable computation by simple bit-shift operations. Our experimental findings have confirmed that this choice of half-step size consistently delivers the most robust performance, surpassing other special values designed for bit shifting, as elaborated in Appendix A.5. The quantization and dequantization procedures for dINT are detailed below:

$$X_{\text{int}} = \begin{cases} c_1, \text{for } \frac{s}{4} < n \le \frac{3s}{4} \\ c_2, \text{for } \frac{-3s}{4} \le n < \frac{-s}{4} \\ \text{clamp}\left(\lceil\frac{X}{s}\rfloor + z, 0, p\right), \text{else} \end{cases} \qquad (5)$$

$$X_q = \begin{cases} \frac{s}{2}, \text{for } X_{\text{int}} = c_1 \\ \frac{-s}{2}, \text{for } X_{\text{int}} = c_2 \\ (X_{\text{int}} - z) \cdot s, \text{else} \end{cases} \qquad (6)$$

where $p$ represents the number of uniform steps, calculated as $p = 2^b - 3$ for a given bit number $b$. The step size $s$ is obtained by dividing the quantization range by $p$, and $z$ is the zero-point for asymmetric quantization. $c_1$ and $c_2$ denote the positive

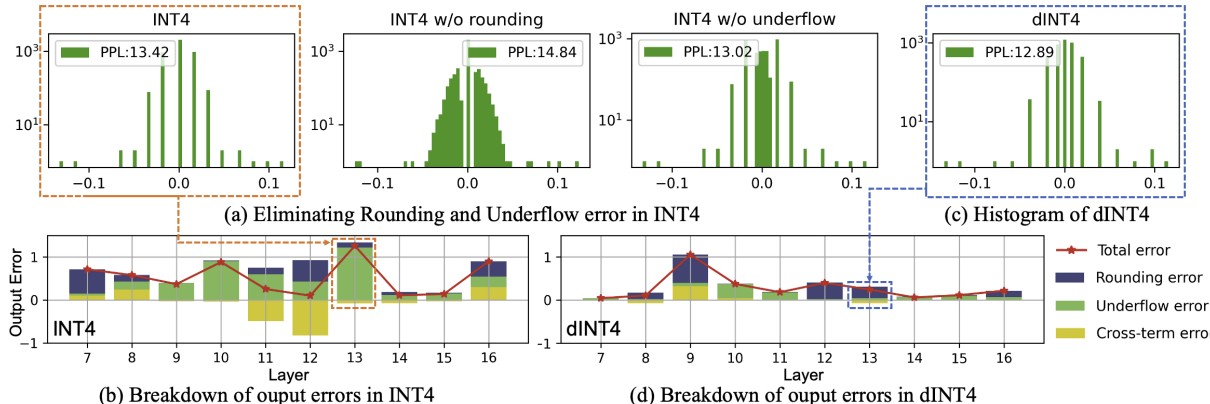

(a) Eliminating Rounding and Underflow error in INT4     (c) Histogram of dINT4

(b) Breakdown of ouput errors in INT4     (d) Breakdown of ouput errors in dINT4

Figure 4: (a) INT4 without rounding sets small values near zero to zero, preserving the rest and causing performance degradation. INT4 without underflow preserves only values near zero, improving performance. (b) Impact of underflow error and rounding error on the output error. Significant impact of underflow error on the output error in INT4. (c) Proposed dINT4 preserves two small values near zero, preventing performance degradation. (d) Using the proposed dINT4 to reduce underflow error leads to a significant reduction in output error.

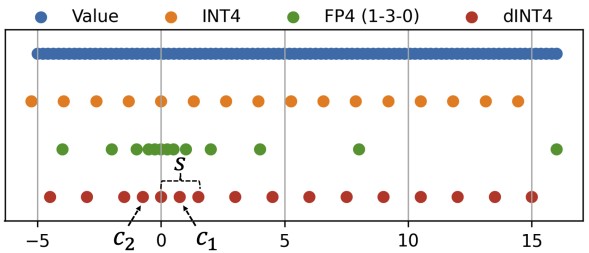

Figure 5: (Blue) Values to be quantized. (Orange) INT4 quantized values, evenly spaced. (Green) FP4 quantized values, dense resolution for small values but coarse resolution for large magnitudes. (Red) Proposed dINT4 format, balanced quantization range with a separate special value for small values.

and negative special values in dINT that represent small magnitudes. These values are encoded with distinct bits, analogous to encoding *inf* or *NaN*. During dequantization, if the value corresponds to $c_1$ or $c_2$, it is represented as a special value; otherwise, dequantization proceeds as in standard integer formats. Fig. 5 shows that dINT4 strikes a balance between INT4, which has uniform dynamic range coverage but underflow issues, and FP4, which densely represents small values to avoid underflow but coarsely covers the dynamic range.

### 4.3 Advantages

Fig. 4(d) showcases the benefits of dINT in reducing output quantization error. By plotting each term of Eq. 4, we observe that dINT primarily mitigates underflow error, which substantially lowers the output error. Although in instances like layer 9, the output error slightly increases due to a widened step size causing a rise in rounding error, the magnitude of this increment is minimal. On the whole,

| HW Performance | MAC Input Formats | |
|---|---|---|
| Precision | INT8 × INT8 | dINT4 × INT8 (Savings) |
| Area ($\mu m^2$) | 86.33 | 44.57 (1.93×) |
| Power (mW) | 0.1595 | 0.0624 (2.56×) |

Table 2: Evaluation of hardware performance of MAC units (7nm, 1GHz).

dINT is effective in most scenarios. Furthermore, we design and synthesize a MAC unit using dINT and compare it to a traditional 8-bit integer MAC unit using Synopsys Design Compiler and a commercial 7nm technology (1GHz) for area efficiency evaluation. As shown in Table 2, dINT achieves 1.93× and 2.56× savings in area and power consumption, respectively. This underscores dINT's effectiveness in tackling underflow issues with minimal output errors and its hardware implementation efficiency.

## 5 Experimental Results

### 5.1 Experimental Settings

In our experimental settings, we implement a comprehensive evaluation to assess the effectiveness of our AQAS, SLAC, and dINT4 techniques in LLMs. This involves conducting quantized inference with 8-bit activations and 4-bit weights across a spectrum of tasks, encompassing language modeling, reasoning, and the MMLU benchmark. To enhance both computational and memory efficiency in activation quantization, we broaden our approach to incorporate the quantization of "*Value* (for attention map calculation)", which are

| Weight Scaling | OPTQ | A-bits | W/V-bits | OPT Family | | | | | | LLaMA Family | | |
|---|---|---|---|---|---|---|---|---|---|---|---|---|
| | | | | 125M | 1.3B | 2.7B | 6.7B | 13B | 30B | 7B | 13B | 30B |
| Baseline | | | | 31.95 | 16.41 | 14.32 | 12.29 | 11.50 | 10.67 | 5.68 | 5.09 | 4.10 |
| - | - | | | 42.78 | 35.17 | 24.03 | 15.52 | 36.43 | 150.94 | 6.87 | 6.14 | 5.52 |
| | ✓ | | | 36.04 | 19.35 | 15.61 | 15.16 | 15.82 | 25.23 | 7.57 | 5.85 | 5.13 |
| SQ | - | INT8 | INT4 | 41.31 | 23.48 | 33.86 | 1,596.83 | 897.25 | 19.43 | 7.08 | 6.27 | 6.17 |
| | ✓ | | | 36.22 | 17.70 | 15.22 | **12.82** | **11.93** | 11.06 | 8.26 | 5.91 | 5.11 |
| AWQ | - | | | 44.17 | 25.04 | 15.88 | 322.85 | 670.00 | 4,246.25 | 6.75 | 5.84 | 5.26 |
| | ✓ | | | 39.10 | 19.35 | 16.00 | 432.74 | 183.83 | 4,848.16 | 6.52 | 5.78 | 5.02 |
| AQAS | - | | | 36.57 | 17.68 | 15.34 | 13.42 | 12.19 | 11.08 | 6.69 | 5.81 | 5.14 |
| | ✓ | | | 35.62 | 17.48 | 15.08 | 12.97 | 12.08 | **11.04** | 6.60 | 5.71 | 5.07 |
| | ✓ | | dINT4 | **34.92** | **17.28** | **15.03** | 12.89 | 11.98 | 11.04 | **6.48** | **5.67** | **4.72** |

Table 3: PPL results of W4A8V4 (Weight-4bit, Activation-8bit, *Value*-4bit) at standard language modeling evaluation with OPT and LLaMA family models, applying OPTQ (Frantar et al., 2023) with various weight scaling techniques and two numerical formats.

specifically cached to expedite the inference stage during generation (Kwon et al., 2023). We compare our methods against baseline techniques, including weight scaling of SQ (Xiao et al., 2022), AWQ (Lin et al., 2023), and weight update based method, OPTQ (Frantar et al., 2023). Task details, models, calibration methods, and quantization techniques used in the experiments are outlined in Appendix A.1, and an ablation study exploring aspects such as reducing precision to 3-bit, weight-only quantization with dINT, and other 4-bit formats is detailed in Appendix A.6.

## 5.2 Evaluation on Language Modeling Task

We first evaluate perplexity (PPL) as the language modeling performance for various PTQ methods. Table 3 presents W4A8 quantization results with different PTQ combinations. For OPT models, OPTQ generally reduces perplexity. However, when combined with weight scaling methods like SQ and AWQ for 4-bit weight quantization, there's a significant accuracy drop (i.e., spikes in PPL), which OPTQ cannot fully mitigate in most OPT models (except 6.7B and 13B). AQAS effectively curtails the accuracy drop of 4-bit weight quantization, and combining it with OPTQ enhances accuracy. Utilizing dINT4 for 4-bit quantization further lowers perplexity, maintaining a gap of less than 1.0 compared to the full-precision baseline, with the exception of OPT-125M, which is sensitive to quantization. For LLaMA models, OPTQ with 4-bit weight quantization raises perplexity due to increased activation diversity, as discussed in Sec. 3.3. Weight scaling methods like AWQ and AQAS aid in performance recovery, and dINT4

| Weight Scaling | OPTQ | W/V-bits | OPT | | LLaMA | |
|---|---|---|---|---|---|---|
| | | | 6.7B | 13B | 7B | 13B |
| Baseline | | | 69.11 | 69.38 | 70.92 | 74.48 |
| - | - | INT4 | 63.23 | 48.67 | 69.37 | 72.22 |
| | ✓ | | 64.42 | 56.62 | 64.94 | 71.69 |
| SQ | ✓ | | 67.80 | 68.96 | 68.23 | 72.63 |
| AQAS | ✓ | | 67.79 | 68.80 | 69.04 | 73.50 |
| | ✓ | dINT4 | **68.19** | **68.96** | 68.74 | 73.31 |
| AQAS* | ✓ | INT4 | 67.48 | 68.24 | 70.48 | 73.50 |
| | ✓ | dINT4 | 67.92 | 68.80 | **71.01** | **73.64** |

Table 4: Average accuracy for CommonSense QA (CSQA) tasks including PIQA, Winogrande, and ARC_easy. AQAS* denotes the AQAS method with the SLAC approach (calibrating the dataset's sequence length to 128).

further minimizes perplexity, keeping it within 1.0 of the baseline. We detail the results of applying our proposed AQAS and dINT4 strategies to models with over 60 billion parameters, specifically OPT-66B and LLaMA-65B, in Appendix A.2.

## 5.3 Evaluation on Zero-shot Reasoning Tasks

We carry out experiments for the zero-shot CommonSense QA (CSQA) (Bisk et al., 2019; Sakaguchi et al., 2019; Clark et al., 2018) benchmark by comparing different quantization options, akin to prior tests. As noted in Sec. 3.3, LLaMA models undergo performance decline with OPTQ without weight scaling, whereas OPT models, less affected by activation diversity, show performance gains using OPTQ even without scaling. Among weight scaling techniques, AQAS exhibits superior performance, and employing the dINT4 format further enhances results.

| Weight Scaling | OPTQ | A-bits | W/V-bits | LLaMA Family | | | |
|---|---|---|---|---|---|---|---|
| | | | | 7B | 13B | 30B | 65B |
| Baseline | | | | 35.20 | 47.15 | 58.50 | 63.60 |
| - | - | INT8 | INT4 | 28.05 | 40.82 | 48.40 | 57.20 |
| | ✓ | | | 27.05 | 42.95 | 53.30 | 58.50 |
| SQ | ✓ | | | 29.32 | 43.12 | 52.83 | 59.30 |
| AQAS | ✓ | | | _30.81_ | _44.23_ | _53.67_ | _59.60_ |
| | ✓ | | dINT4 | **31.00** | **44.73** | **55.50** | **61.40** |

Table 5: Average MMLU accuracy. The detailed accuracy for each item can be found in Table 7.

Due to the shorter input sentences in zero-shot CSQA compared to the default OPTQ calibration dataset, employing SLAC, which considers the LLaMA models' activation diversity based on sequence length, improves performance for both INT4 and dINT4 formats. However, aligning the calibration length with the target task's sequence length for the OPT models does not result in significant improvements. This can be attributed to the OPT models' lower sensitivity to weight updates due to activation diversity during the calibration process, as discussed in Section 3.3, which differs from the behavior of the LLaMA models.

As a result, we attain performance within 1% of full precision for both OPT and LLaMA models using 8-bit activation and 4-bit weight, notably achieving full precision-equivalent performance in LLaMA-7B by comprehensively accounting for the model's activation characteristics.

### 5.4 Evaluation on In-Context Learning Tasks

We evaluate the MMLU benchmark on several options that exhibited strong performance in language modeling. To assess the efficacy of our proposed method in in-context learning, we conduct 5-shot inference. Given that OPT models are deemed unsuitable for the MMLU benchmark (Lin et al., 2023), we restrict the experiments to LLaMA models. Consistent with language modeling results, AQAS, accounting for both weight and activation quantization errors, delivers the best performance. Moreover, effectively managing underflow error bolsters performance across all models, with a notable 2% performance enhancement observed in the LLaMA-30B model. To evaluate the efficacy of our approach on large-scale models, we further expand the experiment to LLaMA-65B. The results demonstrate that dINT4 significantly enhances MMLU accuracy by conserving small-magnitude values. Detailed results for each category within MMLU are provided in the Appendix A.3.

## 6 Conclusion

We address Post-training Quantization (PTQ) in Large Language Models (LLMs), specifically targeting 4-bit weight and 8-bit activation (W4A8) quantization to boost computational efficiency. We present Activation-Quantization-Aware Scaling (AQAS) and Sequence-Length-Aware Calibration (SLAC), refining PTQ by taking into account weights and activations, and aligning sequence lengths. To combat the underflow issue in W4A8 quantization, where small magnitudes are rounded down to zero, we introduce dINT, a hybrid format blending integer and denormal representations. Through extensive evaluations on LLMs such as OPT and LLaMA, we demonstrate marked improvements in task accuracy and adaptability. Additionally, with the development of MAC units compatible with dINT, we achieve a twofold increase in hardware efficiency.

## 7 Limitation

We conducted a thorough analysis of model-specific characteristics in LLMs and identified limitations in current PTQ methods. However, further investigation is needed to understand the specific phenomena observed in certain LLM models during the pre-training process. Additionally, exploring more advanced collaborations of PTQ techniques at lower bit precision for weights and activations holds promise for future research.

## Acknowledgement

This work was supported by Institute of Information & communications Technology Planning & Evaluation(IITP) grants funded by the Korea government(MSIT)(2020-0-01305, Development of AI Deep-Learning Processor and Module for 2,000 TFLOPS Server, 2020-0-01297, Development of Ultra-Low Power Deep Learning Processor Technology using Advanced Data Reuse for Edge Applications), the Technology Innovation Program (1415178807, Development of Industrial Intelligent Technology for Manufacturing, Process, and Logistics) funded By the Ministry of Trade, Industry & Energy(MOTIE, Korea), and the National Research Foundation of Korea (NRF) grant funded by the Korea government(MSIT)(No. 2021R1A2C1013513).

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

# A  Appendix

## A.1  Experimental Details

**Baseline Setup.**  As comparative baselines for weight scaling, we employ SQ (Xiao et al., 2022) as the scale method for 8-bit activation quantization and AWQ (Lin et al., 2023) as the scaling method for 4-bit weight quantization. In terms of weight rounding, we evaluate both options provided by OPTQ (Frantar et al., 2023), which offers additional optimization, and the standard nearest rounding method. As for numerical format, we compare dINT with existing INT4 methods in terms of performance.

**Task and Models.**  We evaluate our proposed approach on various tasks, including language modeling using Wikitext (Merity et al., 2016)), CSQA (PIQA (Bisk et al., 2019), WinoGrande (Sakaguchi et al., 2019), and ARC easy (Clark et al., 2018)), as well as MMLU (Hendrycks et al., 2021) benchmarks. For benchmark models, we assess a range of options across OPT and LLaMA models, ranging from 125M to 66B, which are widely used LLMs.

**Quantization Settings.**  We apply quantization to both the weights and activations of all matrix multiplications in the Decoder layer. We conduct our experiments by implementing the quantizer within the PyTorch framework. For activations, except for *Value*, we apply 8-bit quantization, while for memory-intensive components such as weights and *Value*, we utilize 4-bit quantization. Similar to commonly used methods for LLM quantization (Dettmers et al., 2022; Yao et al., 2022), we apply token-wise quantization for activations, and output channel-wise quantization for weights. For *Value*, we apply channel-wise quantization, taking into account the dimensions where partial-sum accumulation occurs when multiplied by the self-attention map. We apply Min-Max asymmetric quantization determine the step size and the zero-point for both activation and weight.

**Calibration Settings.**  During the calibration process to find the weight scale, we follow the calibration setting from the AWQ repository[1]. For the attention operation, we adjust the cali-

| Wikitext PPL | OPT-66B | LLaMA-65B |
|---|---|---|
| FP16 baseline | 10.09 | 3.53 |
| INT4 | 3,222.02 | 4.80 |
| AQAS + dINT4 + OPTQ | 10.32 | 4.13 |

Table 6: Wikitext perplexity for W4A8V4 inference on OPT-66B and LLaMA-65B.

bration process by modifying the objective of Eq. 1 to minimize the distortion of the attention output. We use a randomly extracted dataset from Pile (Gao et al., 2020) for AWQ (Lin et al., 2023), SmoothQuant (Xiao et al., 2022), and AQAS methods. When calibrating weights with OPTQ, we follow the baseline calibration setting provided in the OPTQ repository[2]. We use a subset of the C4 dataset, randomly selecting 128 samples with a sequence length of 2048.

## A.2  Language modeling in >60B Models

Table 6 To assess the effectiveness of our approach on large models, we conduct language modeling experiments on OPT-66B and LLaMA-65B, with the objective of determining whether our method performs well even on models with over 60 billion parameters. demonstrates that proposed scaling method and numerical format can significantly reduce perplextiy in language modeling task.

## A.3  Few-shot MMLU Benchmarks

The results for the each category in the 5-shot MMLU benchmark for LLaMA models are displayed in Table 7. As demonstrated in Table 7, AQAS exhibits higher accuracy compared to other scaling methods, emphasizing the importance of considering both weight and activation quantization. Furthermore, it's noteworthy that the use of dINT, which effectively mitigates underflow, achieves the highest accuracy.

## A.4  Finding Scales for AQAS

To automatically determine the channel-wise scale factor in AQAS, it is necessary to select representative values for both activation and weight channels. In SQ (Xiao et al., 2022), the maximum magnitude was used as the criterion, while in AWQ (Lin et al., 2023), the absolute mean value was used to explore the scale factor. As shown in Table 8, we explore both cases and found that selecting the

---

[1]https://github.com/mit-han-lab/llm-awq

[2]https://github.com/IST-DASLab/gptq

| LLaMA 7B | | | | | | | | |
|---|---|---|---|---|---|---|---|---|
| Scaling | OPTQ | A-bits | W/V-bits | Hums | STEM | Social | Other | Avg. |
| Baseline | | | | 33.90 | 30.50 | 38.20 | 38.20 | 35.20 |
| - | - | INT8 | INT4 | 28.65 | 25.75 | 26.13 | 31.15 | 28.05 |
| - | ✓ | INT8 | INT4 | 26.99 | 24.88 | 25.71 | 30.41 | 27.05 |
| SQ | ✓ | INT8 | INT4 | 27.80 | 29.66 | 27.40 | 33.04 | 29.32 |
| AQAS | ✓ | | INT4 | 29.61 | 29.85 | 30.65 | 33.62 | 30.81 |
| AQAS | ✓ | | dINT4 | 29.67 | 29.26 | 32.24 | 33.37 | **31.00** |

| LLaMA 13B | | | | | | | | |
|---|---|---|---|---|---|---|---|---|
| Scaling | OPTQ | A-bits | W/V-bits | Hums | STEM | Social | Other | Avg. |
| Baseline | | | | 44.80 | 36.40 | 54.20 | 53.20 | 47.15 |
| - | - | INT8 | INT4 | 36.96 | 35.16 | 45.56 | 47.19 | 40.82 |
| - | ✓ | INT8 | INT4 | 40.38 | 34.10 | 48.94 | 49.23 | 42.95 |
| SQ | ✓ | INT8 | INT4 | 40.96 | 34.26 | 48.72 | 49.20 | 43.12 |
| AQAS | ✓ | | INT4 | 40.74 | 35.02 | 51.51 | 50.96 | 44.23 |
| AQAS | ✓ | | dINT4 | 42.64 | 35.79 | 50.83 | 50.31 | **44.73** |

| LLaMA 30B | | | | | | | | |
|---|---|---|---|---|---|---|---|---|
| Scaling | OPTQ | A-bits | W/V-bits | Hums | STEM | Social | Other | Avg. |
| Baseline | | | | 56.40 | 46.70 | 67.30 | 63.60 | 58.50 |
| - | - | INT8 | INT4 | 46.00 | 39.40 | 54.50 | 54.50 | 48.40 |
| - | ✓ | INT8 | INT4 | 50.20 | 42.60 | 61.90 | 59.70 | 53.30 |
| SQ | ✓ | INT8 | INT4 | 49.65 | 43.31 | 59.86 | 59.65 | 52.83 |
| AQAS | ✓ | | INT4 | 51.75 | 43.21 | 60.68 | 59.53 | 53.67 |
| AQAS | ✓ | | dINT4 | 52.90 | 44.00 | 65.00 | 61.10 | **55.50** |

| LLaMA 65B | | | | | | | | |
|---|---|---|---|---|---|---|---|---|
| Scaling | OPTQ | A-bits | W/V-bits | Hums | STEM | Social | Other | Avg. |
| Baseline | | | | 61.90 | 52.10 | 73.40 | 67.60 | 63.60 |
| - | - | INT8 | INT4 | 54.10 | 46.40 | 66.90 | 62.50 | 57.20 |
| - | ✓ | INT8 | INT4 | 56.00 | 47.80 | 67.20 | 63.90 | 58.50 |
| SQ | ✓ | INT8 | INT4 | 57.70 | 47.50 | 67.90 | 64.30 | 59.30 |
| AQAS | ✓ | | INT4 | 57.20 | 47.80 | 69.50 | 64.80 | 59.60 |
| AQAS | ✓ | | dINT4 | 59.50 | 50.40 | 70.70 | 65.80 | **61.40** |

Table 7: MMLU accuracy on LLaMA models.

maximum magnitude as the representative value often yielded better performance. Similar to previous research (Lin et al., 2023), we use a grid search to find the appropriate scale, and after determining the scale factor, we make adjustments by additionally clipping the weights.

## A.5 Sweep of the Special Value in dINT

The dINT format defines the special value $c$ as half of the step size $s$. If we change this value, the magnitude of the state representing small values will differ. We conduct additional sweeps with different power-of-two values (e.g., 0.25, 0.125) to observe the impact in Table 9. In most cases, setting $c$ to 0.25 times the $s$ proves to be a good choice, but in the case of the OPT-125M model, it shows a significant increase in perplexity. To select a value that generally works well, we set $c$ to be half of $s$.

## A.6 Ablation Study

**Reducing Precision to 3 Bits.** To achieve additional memory savings, we conduct experiments

| Wikitext | | | | | | | | |
|---|---|---|---|---|---|---|---|---|
| AQAS | OPTQ | OPT | | | | | | LLaMA |
| | | 125M | 1.3B | 2.7B | 6.7B | 13B | 30B | 7B |
| Baseline | | 31.95 | 16.41 | 14.32 | 12.29 | 11.50 | 10.67 | 5.68 |
| Mean | - | 37.46 | 18.16 | 15.43 | 13.08 | 12.11 | 11.06 | 6.72 |
| Mean | ✓ | 36.07 | 17.50 | 15.19 | 12.99 | **12.07** | **11.02** | 6.67 |
| Max | - | 36.57 | 17.68 | 15.34 | 13.42 | 12.19 | 11.08 | 6.69 |
| Max | ✓ | **35.62** | **17.48** | **15.08** | **12.97** | 12.08 | 11.04 | **6.60** |

| PIQA | | | | | | | | |
|---|---|---|---|---|---|---|---|---|
| AQAS | OPTQ | OPT | | | | | | LLaMA |
| | | 125M | 1.3B | 2.7B | 6.7B | 13B | 30B | 7B |
| Baseline | | 63.00 | 71.71 | 73.78 | 76.28 | 75.90 | 77.58 | 78.35 |
| Mean | - | 61.86 | 70.78 | 72.91 | **75.57** | 74.86 | 76.66 | **77.42** |
| Mean | ✓ | 61.81 | **71.22** | 72.85 | 75.14 | 74.97 | 77.20 | 77.37 |
| Max | - | 61.21 | 70.51 | 72.69 | 74.27 | 75.14 | 77.15 | 77.26 |
| Max | ✓ | **62.30** | 71.11 | **73.56** | 75.08 | **75.84** | **77.48** | 76.28 |

| WinoGrande | | | | | | | | |
|---|---|---|---|---|---|---|---|---|
| AQAS | OPTQ | OPT | | | | | | LLaMA |
| | | 125M | 1.3B | 2.7B | 6.7B | 13B | 30B | 7B |
| Baseline | | 50.28 | 59.51 | 61.01 | 65.43 | 65.11 | 73.01 | 67.09 |
| Mean | - | 51.54 | 58.88 | 60.14 | 65.59 | **65.51** | 67.56 | 64.96 |
| Mean | ✓ | 49.96 | 57.62 | **61.64** | **65.82** | 64.88 | 68.03 | 64.80 |
| Max | - | **52.41** | **60.62** | **61.64** | 65.27 | 64.88 | 67.40 | 63.77 |
| Max | ✓ | 49.72 | 58.17 | 60.14 | 63.77 | 65.27 | **68.11** | **65.59** |

Table 8: Comparing the performance of AQAS when exploring channel-wise quantization using the criteria of absolute mean and max values.

| $c_1 / s$ | OPT | | | | LLaMA 7B | Avg. |
|---|---|---|---|---|---|---|
| | 125M | 1.3B | 2.7B | 6.7B | | |
| Baseline | 31.95 | 16.41 | 14.32 | 12.29 | 5.68 | 16.13 |
| 0.50 | 34.92 | 17.28 | 15.03 | 12.89 | 6.48 | **17.32** |
| 0.25 | 35.84 | 17.22 | 14.96 | 12.83 | 6.28 | 17.43 |
| 0.125 | 35.20 | 17.23 | 14.97 | 12.84 | 6.32 | **17.32** |

Table 9: dINT4's special value sweep, W4A8V4 inference with AQAS+OPTQ. Where $c_1$ is the positive special value, and $s$ is the step size.

| Weight scaling | OPTQ | W/V-bits | OPT | | | LLaMA |
|---|---|---|---|---|---|---|
| | | | 125M | 2.7B | 6.7B | 7B |
| Baseline | | | 31.95 | 14.32 | 12.29 | 5.68 |
| - | - | INT3 | 1.7e3 | 4.3e4 | 1.2e4 | 94.97 |
| - | - | dINT3 | 127.94 | 8.7e3 | 55.24 | 10.99 |
| AQAS | ✓ | INT3 | 54.84 | 36.38 | 69.45 | 24.85 |
| AQAS | ✓ | dINT3 | **46.34** | **20.67** | **17.42** | **10.04** |

Table 10: Perplexity is assessed using standard language modeling on the Wikitext dataset, with activations quantized to 8 bits and weights and values to 3 bits. We employ AQAS and OPTQ and compare the performance of INT3 and dINT3. Notably, dINT3 considerably reduces performance degradation.

in which we retain 8-bit activation while reducing weight and *Value* precision to 3 bits. As shown in Table 10, as bit precision decreases and the impact of underflow becomes more significant, the effectiveness of dINT becomes more pronounced.

| Precision | Format | Method | OPT | | | | |
|---|---|---|---|---|---|---|---|
| | | | 125M | 1.3B | 2.7B | 6.7B | 13B |
| FP16 baseline | | | 31.95 | 16.41 | 14.32 | 12.29 | 11.50 |
| W3A16 g=128 | INT3 | RTN | 58.37 | 195.10 | 499.39 | 39.18 | 29.37 |
| | | OPTQ | 41.93 | 18.53 | 15.79 | 13.13 | 12.01 |
| | | AWQ | 41.45 | 18.56 | 15.63 | 12.99 | 12.03 |
| | dINT3 | RTN | 54.53 | 21.70 | 24.15 | 15.80 | 13.14 |
| | | OPTQ | 39.60 | **18.26** | 15.52 | 12.99 | **11.91** |
| | | AWQ | **39.46** | 18.29 | **15.47** | **12.97** | 12.03 |

Table 11: Performance comparison of W3A16 inference results with various state-of-the-art methods when applying group-wise quantization (group size: 128).

| Precision | Format | Method | OPT | | | | |
|---|---|---|---|---|---|---|---|
| | | | 125M | 1.3B | 2.7B | 6.7B | 13B |
| FP16 baseline | | | 31.95 | 16.41 | 14.32 | 12.29 | 11.50 |
| W4A16 g=128 | INT4 | RTN | 35.52 | 17.69 | 15.12 | 13.02 | 11.89 |
| | | OPTQ | 34.23 | 16.92 | 14.69 | 12.51 | 11.60 |
| | | AWQ | 33.96 | 16.85 | 14.61 | **12.44** | 11.60 |
| | FP4 | RTN | 39.13 | 18.25 | 15.54 | 13.35 | 12.14 |
| | | OPTQ | 37.42 | 18.06 | 15.19 | 12.84 | 11.91 |
| | dINT4 | RTN | 34.40 | 17.06 | 14.83 | 12.75 | 11.79 |
| | | OPTQ | 34.04 | 16.82 | 14.64 | 12.47 | **11.59** |
| | | AWQ | **33.66** | **16.78** | **14.56** | **12.44** | 11.61 |

Table 12: Performance comparison of W4A16 inference results with various state-of-the-art methods when applying group-wise quantization (group size: 128).

By solely changing the numerical format without applying weight scaling, we are able to significantly reduce the perplexity of the LLaMA-7B model from 94.97 to 10.99. This underscores the influence of underflow on model performance.

**Weight-Only Quantization Method with dINT.** dINT, as a numerical format, can be integrated with existing PTQ methods. We combine the dINT format with state-of-the-art PTQ methods for LLMs, namely OPTQ and AWQ, and compare their performance with the integer format. As shown in Table 11 and Table 12, dINT outperforms the traditional integer format in both 3-bit and 4-bit quantization. This indicates that underflow significantly affects the performance of weight quantization in LLMs.

**Other 4-Bit Formats.** To compare the performance of 4-bit quantization formats, we evaluate performance by applying integer, floating-point, and dINT4 to the weights, without considering activation quantization. We employ a 4-bit floating-point (FP4), consisting of a single sign bit and three exponent bits. While alternative configurations with different exponent and mantissa bits are available, we experimentally determine the necessity of a 3-bit exponent for the FP4. Additional

| Model | Precision | W4 format | Wikitext | PIQA | MMLU |
|---|---|---|---|---|---|
| LLaMA-7B | | FP16 baseline | 5.68 | 78.29 | 35.20 |
| | W4A16 | FP4 (1-1-2) | 165582.55 | 51.69 | 26.88 |
| | | FP4 (1-2-1) | 26.52 | 62.84 | 27.31 |
| | | FP4 (1-3-0) | 6.30 | 76.77 | 31.46 |
| | | dINT4 | **6.07** | **77.91** | **32.53** |
| LLaMA-13B | | FP16 baseline | 5.09 | 78.78 | 47.15 |
| | W4A16 | FP4 (1-1-2) | 74763.98 | 52.29 | 24.72 |
| | | FP4 (1-2-1) | 7.95 | 74.65 | 31.54 |
| | | FP4 (1-3-0) | 5.56 | 78.62 | 40.76 |
| | | dINT4 | **5.38** | **79.05** | **44.35** |
| LLaMA-30B | | FP16 baseline | 4.10 | 80.96 | 58.50 |
| | W4A16 | FP4 (1-1-2) | 34027.07 | 51.52 | 25.32 |
| | | FP4 (1-2-1) | 9.10 | 71.22 | 32.05 |
| | | FP4 (1-3-0) | 4.57 | 79.71 | 53.50 |
| | | dINT4 | **4.36** | **80.41** | **55.87** |

Table 13: Experiments on various configurations of 4-bit floating-point: FP4 (1-$e$-$m$) represents floating-point format with a 1-bit sign bit, $e$-bit exponent, and $m$-bit mantissa. We conduct experiments on Wikitext PPL, PIQA accuracy, and MMLU average accuracy. Among FP4 configurations, a 3-bit exponent exhibits the best performance, while dINT surpassing it.

| Precision | Format | OPTQ | OPT | | | | |
|---|---|---|---|---|---|---|---|
| | | | 125M | 1.3B | 2.7B | 6.7B | 13B |
| FP16 baseline | | | 31.95 | 16.41 | 14.32 | 12.29 | 11.50 |
| W4A16 | INT4 | - | 43.15 | 29.90 | 19.70 | 14.18 | 12.89 |
| | | ✓ | 36.29 | 17.68 | 15.15 | 12.88 | 11.73 |
| | FP4 | - | 42.17 | 18.52 | 16.01 | 13.38 | 12.33 |
| | | ✓ | 37.52 | 18.31 | 15.39 | 13.18 | 11.89 |
| | dINT4 | - | 37.16 | 18.10 | 15.53 | 13.75 | 12.06 |
| | | ✓ | **35.10** | **17.30** | **14.94** | **12.65** | **11.68** |

Table 14: Comparing the performance of various formats in weight-only quantization for the language modeling task, using both dINT and OPTQ together shows the best performance.

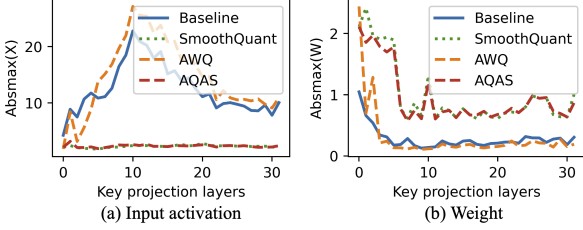

Figure 6: The variation in the absolute max values of weights and activations when applying weight scaling in LLaMA-7B.

details can be found in Table 13. As shown in Table 14, FP4 achieves some performance improvement compared to uniform quantization due to its wider dynamic range. However, dINT4 outperforms the other two formats by effectively representing a wide range of values with uniform intervals while accurately representing small values. It demonstrates better performance and good compatibility with existing optimization techniques such as OPTQ.