# OpenReview forum: "Enhancing Computation Efficiency in Large Language Models through Weight and Activation Quantization"
_EMNLP/2023/Conference — EMNLP 2023 Main_

### Official Review · Reviewer_XAcF · 2023-08-04

**Soundness:** 3

**Excitement:**

3: Ambivalent: It has merits (e.g., it reports state-of-the-art results, the idea is nice), but there are key weaknesses (e.g., it describes incremental work), and it can significantly benefit from another round of revision. However, I won't object to accepting it if my co-reviewers champion it.

**Paper Topic And Main Contributions:**

The paper addresses the quantization problem in Large Language Models by targeting the 4-bit weight and 8-bit activation (W4A8) quantization. It proposes activation-quantization-aware scaling (AQAS) to jointly quantize the weight and activation. Additionally, sequence-length-aware calibration (SLAC) is introduced to match sequence lengths and align calibration data ranges with target tasks. The dINT method is utilized to denormalize the representation.

**Questions For The Authors:**

1.	Why wasn't the AWQ method compared in Table 4 and 5?
2.	SLAC is conspicuously absent from most experiments without being thoroughly analyzed. However, it does provide some assistance in improving results, as evidenced in table 4 during the LLaMA experiments. The reason behind excluding SLAC from other experiment sets remains unclear. Does this imply that SLAC generally does not contribute to accuracy improvement in most cases?
3.	Although the paper highlights a twofold increase in hardware efficiency, the provided experiments primarily focus on accuracy. Where can we find evidence or information demonstrating the improved hardware efficiency?


**Reasons To Accept:**

The paper is mostly well-written and easy to follow. The frame is clear and easy to understand. The methods are described with enough detail, and the results are well presented and discussed. The paper analyzes in great detail the changes of weight and activation in each layer of the LLMs during the quantization process and gives solutions for each of the problem.

**Reasons To Reject:**

1.	In Section 3.1, there are inconsistencies in the description and arrangement of Figure 1(b). The description states that the min-max range of activation should be at the top and weight at the bottom, but the figure is presented in a left-right format. Additionally, the legend in Figure 1(b) only provides two descriptions, despite there being four lines in the graph. It can be inferred that the lower part represents the min value, and the upper part represents the max value, but it would be advisable to visually distinguish these parts using different colors or markers for improved clarity.
2.	Upon analyzing Figure 2, I perceive a limited distinction between SmoothQuant and AQAS. This similarity becomes more pronounced as the numbers increase, ultimately resulting in a diminishing difference and even intersecting towards the end.
3.	The paper lacks detailed information and fails to provide a comprehensive description of the SLAC method. For instance, it does not present the formula or describe the process involved in implementing SLAC.
4.	The paper highlights an achievement of a twofold increase in hardware efficiency. However, it fails to include any experiments specifically demonstrating this improvement in hardware efficiency. Rather, the paper solely focuses on presenting accuracy results without a corresponding set of experiments showcasing the enhancement in hardware efficiency.
5.	Why wasn't the AWQ method compared in Table 4 and 5?
6.	SLAC is noticeably absent from the majority of experiments and lacks thorough analysis. Its presence is only observed in table 4 during the LLaMA experiments, where SLAC shows some advantages. The reasoning behind excluding SLAC from other sets of experiments is unclear. It is recommended to include more analysis and comparisons of SLAC to provide a comprehensive evaluation.


**Reproducibility:**

3: Could reproduce the results with some difficulty. The settings of parameters are underspecified or subjectively determined; the training/evaluation data are not widely available.

**Reviewer Confidence:**

3: Pretty sure, but there's a chance I missed something. Although I have a good feel for this area in general, I did not carefully check the paper's details, e.g., the math, experimental design, or novelty.

---

> ### Author Rebuttal · Authors · 2023-08-29
>
> We sincerely appreciate the insightful feedback provided by the reviewers. In response to the raised questions, our answers are as follows.
>
> > **Reasons To Reject 1.** In Section 3.1, there are inconsistencies in the description and arrangement of Figure 1(b). The description states that the min-max range of activation should be at the top and weight at the bottom, but the figure is presented in a left-right format. Additionally, the legend in Figure 1(b) only provides two descriptions, despite there being four lines in the graph. It can be inferred that the lower part represents the min value, and the upper part represents the max value, but it would be advisable to visually distinguish these parts using different colors or markers for improved clarity.
>
> Thank you for your detailed comments. We apologize for any confusion caused by the incorrect placement and description of Figure 1 (b). In line with the reviewer’s suggestions, we will align the description and arrangement of Figure 1 (b) for clarity. Additionally, we will update in the final manuscript the legend to accurately represent all four lines, improving the visualization of four lines in the graph.
>
> &nbsp;
>
> > **Reasons To Reject 2**. Upon analyzing Figure 2, I perceive a limited distinction between SmoothQuant and AQAS. This similarity becomes more pronounced as the numbers increase, ultimately resulting in a diminishing difference and even intersecting towards the end.
>
> Thank you for your detailed question regarding the distinctive differences between SmoothQuant (SQ) and AQAS. We aim to elucidate the differences between SQ and AQAS by highlighting that they utilize different **scale values**, leading to significant discrepancies in the **quantization layer output error**, and these differences consequently impact **text generation performance.**
>
> To illustrate the impact of different scaling methods, we measured the scale values and the subsequent output quantization errors (via SQNR) for each layer in both SQ and AQAS, as presented in Table R1. Notably, in Layers 5 and 10, AQAS employs smaller scale values than SQ, corroborating the trends in absmax of weight seen in Figure 2. **These contrasting scale values lead to substantially higher output quantization errors** in the initial layers for SQ compared to AQAS. Even though similar scale values emerge in later layers, these variations result in markedly different text generation performance, as evidenced by the PPL comparisons between SQ and AQAS.
>
> OPT-6.7B | Layer | 5 | 10 | 20 | 30 | W4A8 PPL
> -- | -- | -- | -- | -- | -- | --
> SQ | Scale | 22.55 | 22.39 | 25.92 | 31.53 | 1586.83
> SQ | SQNR | 12.51 | 4.29 | 11.47 | 9.71 | -
> AQAS | Scale | 13.44 | 14.91 | 21.69 | 28.45 | **13.42**
> AQAS | SQNR | **16.76** | **5.91** | **11.90** | **9.93** | -
>
> Table R1. Comparison of scale for outlier channel and per-layer SQNR between SQ and AQAS in OPT-6.7B and wikitext-2 PPL results of W4A8 PTQ OPT-6.7B (SQNR: signal-to-quantization-noise ratio, higher the better)
>
> To assess the impact of the analys on model accuracy, we conducted a performance comparison between SQ and AQAS on the recently released LLaMA-2 model. As seen in Table R2, AQAS consistently outperforms SQ across various tasks, including wikitext, MMLU, and CSQA, when applied to the LLaMA-2 model. This confirms that AQAS achieves significantly higher performance compared to SQ, reinforcing its robustness and efficacy.
>
> LLaMA-2-7B (W4A8) | WikiText-2 | MMLU | CSQA
> -- | -- | -- | --
> Weight Scaling | PPL | Acc | Acc
> FP | 5.42 | 41.6 | 71.76
> SQ | 6.82 | 38.6 | 68.55
> AQAS | 6.46 | 40.7 | 70.40
> SQ+OPTQ | 6.65 | 38.9 | 69.51
> AQAS+OPTQ | **6.45** | **41.1** | **70.94**
>
> Table R2. Performance comparison between SQ and AQAS across wikitext, MMLU, and CSQA (PIQA, winogrande, ARC_easy) tasks with LLaMA-2-7B.
>
> &nbsp;
>
> >**Reasons To Reject 3.**  The paper lacks detailed information and fails to provide a comprehensive description of the SLAC method. For instance, it does not present the formula or describe the process involved in implementing SLAC.
> **Reasons To Reject 6.** SLAC is noticeably absent from the majority of experiments and lacks thorough analysis. Its presence is only observed in Table 4 during the LLaMA experiments, where SLAC shows some advantages. The reasoning behind excluding SLAC from other sets of experiments is unclear. It is recommended to include more analysis and comparisons of SLAC to provide a comprehensive evaluation.
> **Question 2.** SLAC is conspicuously absent from most experiments without being thoroughly analyzed. However, it does provide some assistance in improving results, as evidenced in Table 4 during the LLaMA experiments. The reason behind excluding SLAC from other experiment sets remains unclear. Does this imply that SLAC generally does not contribute to accuracy improvement in most cases?
>
> Thank you for asking for further clarification on SLAC. SLAC is a method emerged from observation that the mismatch in the sequence length between the calibration and the target tasks results in significant accuracy degradation, which has been neglected for the careful study in previous work. To address the reviewer's questions comprehensively, we have prepared detailed responses with two sections.
>
> **[Motivation and detailed description of SLAC]**
>
> SLAC aligns the sequence length of the application task with that of the PTQ calibration dataset, thereby enhancing the PTQ performance of the target task.
>
> - This approach is motivated by our discovery that variations in activation diversity, depending on the input sequence length, have a significant impact on the PTQ calibration process, as shown in Table 1.
> - To mitigate such impacts, SLAC aligns the expected sequence length in the target task inference phase with the sequence length of the calibration dataset.
> - The specific process of our proposed SLAC method is as follows: 1) Measure the expected sequence length for the target application task during the inference phase. 2) In the OPTQ calibration process, align the sequence length of the calibration dataset with the previously measured sequence length.
> - By following this approach, we can apply task-specific PTQ calibration, thereby enhancing the accuracy of the quantized model's inference under the target task in a robust manner.
>
> **[Experimental results and analysis of SLAC]**
>
> We conducted experiments applying the SLAC method to the MMLU task. Through this experiment, we not only observed performance improvement but also conducted an analysis of SLAC with individual sub-tasks within MMLU. Below is a summary of our experimental findings and analysis.
>
> - In Table R3, **SLAC**: adjusting the sequence length of OPTQ calibration dataset based on the averaged sequence length (698) across every subject's dataset in MMLU, **default**: using a default sequence length (2048) employed in OPTQ. The evaluation was conducted both with and without the AQAS scaling method.
>
>    - In both LLaMA-7B and LLaMA-13B, we observed that the SLAC method improves the performance of MMLU. Notably, the performance enhancement was seen in both OPTQ and AQAS-scaled OPTQ, confirming that the SLAC method can be orthogonally applied to both calibration-based PTQ approaches (OPTQ) and the scaling-based methods combined with OPTQ (AQAS+OPTQ).
>
> - To investigate the effectiveness of the SLAC methodology, we selected a subset of subjects from the MMLU dataset, each characterized by different average sequence lengths. Our findings, summarized in Table R4, compared the accuracy of two PTQ models: SLAC-1978 and SLAC-587. These models were calibrated to the average sequence lengths specific to the History and Chemistry subjects, respectively, both of which are sub-tasks within MMLU.
>
>    - The results revealed a consistent trend: models calibrated to match the average sequence length of each subject yielded higher accuracy. This finding underscores the efficacy of the SLAC methodology, emphasizing that **task-specific customization of calibration sequence length** can lead to improved performance in W4A8 PTQ.
>
> MMLU | Calibration (Seq Len) | LLaMA-7B | LLaMA-13B
> -- | -- | -- | --
> |OPTQ | default (2048) | 27.4 | 43.0
> |  | SLAC (698.1) | **28.8** | **43.7**
> |AQAS+OPTQ | default (2048) | 31.0 | 44.2
> |  | SLAC (698.1) | **31.6** | **44.6**
>
> Table R3. W4A8 MMLU accuracy comparison between default calibration sequence length (2048) and SLAC (698.1) method for total MMLU dataset using OPTQ (+AQAS) PTQ method.
>
> MMLU single subject | Calibration (Seq Len) | History (seq len avg. 1978) | Chemistry (seq len avg. 587)
> -- | -- | -- | --
> | OPTQ | SLAC (1978) | **51.9** | 27.1
> | OPTQ | SLAC (587) | 51.2 | **30.4**
>
> Table R4. Results of varying calibration sequence length across each subject (History, Chemistry) of MMLU with LLaMA-7B (W4A8)
>
> &nbsp;
>
> > **Reasons To Reject 4.** The paper highlights an achievement of a twofold increase in hardware efficiency. However, it fails to include any experiments specifically demonstrating this improvement in hardware efficiency. Rather, the paper solely focuses on presenting accuracy results without a corresponding set of experiments showcasing the enhancement in hardware efficiency.
> **Question 3.** Although the paper highlights a twofold increase in hardware efficiency, the provided experiments primarily focus on accuracy. Where can we find evidence or information demonstrating the improved hardware efficiency?
>
> As highlighted by the reviewer, we have quantified the proposed dINT4xINT8 through a comparative analysis of area and power consumption against the state-of-the-art configuration, INT8xINT8 in Table 2. Our study primarily explored whether the dINT4-based MAC unit could offer performance gains proportionate to the bit-precision reduction over the INT8-based unit. As noted in previous research [R1], lowering bit-precision can proportionally enhance both area and power efficiency, leading to increased throughput (e.g., transitioning from FP8-FP8 to INT4-INT4 boosts throughput 4x, as per [R2]). Table 2 shows that by reducing weight precision from 8-bit to 4-bit, our dINT format yields 2x area and 2.5x power efficiency improvements, underscoring dINT's hardware efficiency to offer scalable performance despite its representation (Eq. 5) distinct to INT.
>
> [R1] M. Horowitz. "Energy table for 45nm process", Stanford VLSI wiki. [Online]. Available: https://sites.google.com/site/seecproject
> [R2] Agrawal et al., “9.1 A 7nm 4-Core AI Chip with 25.6TFLOPS Hybrid FP8 Training, 102.4TOPS INT4 Inference and Workload-Aware Throttling”, ISSCC 2021
>
> &nbsp;
>
> > **Reasons To Reject 5.** Why wasn't the AWQ method compared in Tables 4 and 5?
> **Question 1.** Why wasn't the AWQ method compared in Table 4 and 5?
>
> We apologize for the confusion caused by not reporting the results of the AWQ method in Tables 4 and 5. We compared the performance of the AWQ method with SQ and AQAS methods in W4A8 performance for CSQA (PIQA, ARC_easy) and MMLU evaluation.
>
> - In the case of OPT-13B in Table R5, significant performance degradation is observed with AWQ+OPTQ. This result can be attributed to the AWQ scaling approach, which only considers weight quantization and exacerbates the difficulty of activation quantization in OPT, which has a wide dynamic range compared to LLaMA.
>
> - In the LLaMA model, which has opposite activation-weight dynamic ranges compared to OPT, the challenge of activation quantization is less pronounced, resulting in less performance degradation. However, its performance still falls short compared to AQAS in both CSQA and MMLU benchmarks.
>
> CSQA/MMLU | W/V-format | PIQA | ARC-easy | PIQA | ARC-easy | MMLU-avg
> -- | -- | -- | -- | -- | -- | --
> | scale method |  model  | OPT-13B | OPT-13B | LLaMA-13B | LLaMA-13B | LLaMA-13B
> | - | FP | 75.90 | 67.13 | 78.78 | 74.58 | 47.15
> |- | INT4 | 66.16 | 50.34 | 77.80 | 70.41 | 42.95
> |SQ + OPTQ | INT4 | 75.52 | 66.33 | 78.56 | 71.21 | 43.12
> |AWQ + OPTQ | INT4 | 72.09 | 63.08 | 78.56 | 72.64 | 44.55
> |AQAS + OPTQ | INT4 | **75.84** | 65.28 | **78.84** | **73.23** | 44.23
> |AQAS + OPTQ | dINT4 | 75.41 | **66.12** | 78.02 | 72.77 | **44.73**
>
> Table R5. CSQA (PIQA, ARC_easy) and MMLU evaluation comparison between SQ, AWQ, and AQAS methods in W4A8 precision
>
> Through comparison with AWQ, we can confirm that AQAS consistently delivers **high performance across both OPT and LLaMA models,** demonstrating its robustness.

---

### Official Review · Reviewer_MJRd · 2023-08-04

**Typos Grammar Style And Presentation Improvements:** n/a
**Soundness:** 4

**Excitement:**

3: Ambivalent: It has merits (e.g., it reports state-of-the-art results, the idea is nice), but there are key weaknesses (e.g., it describes incremental work), and it can significantly benefit from another round of revision. However, I won't object to accepting it if my co-reviewers champion it.

**Missing References:**

n/a

**Paper Topic And Main Contributions:**

This paper focuses on using post-training quantization (PTQ) in LLMs to enhance computational efficiency. . The authors conduct study on applying W4A8 PTQ in specific LLM models (OPT and LLaMa) inference.

This main contribution of this paper is that it pushes the limit of LLMs inference compute efficiency. Other contributions include:
1. The authors combine exitsing methods SQ and AWQ to control the distribution of outliers of activation and weight.
2. The authors find there is relation between variation in activation diversity and input sequence lengths in OPT and LLaMa, and propose Sequence-Length-Aware Calibration (SLAC) to mitigate accuracy losses.
3. The authors propose a new data format dINT to reduce underflow in low precision.

**Questions For The Authors:**

What is the unique contribution of this paper compared with similar investigations around W4A8 PTQ for language models: https://openreview.net/pdf?id=tvDRmAxGIjw, and clamping denormal strategy: https://arxiv.org/pdf/1909.13271.pdf?

**Reasons To Accept:**

1. The authors have conducted a large number of tests on LLMs, and show that the W4A8 PTQ works well. The results are solid.

2. The authors have done model analysis and found interesting relations between distribution of outliers of activation and weight, variation in activation diversity and input sequence lengths in OPT and LLaMa.

**Reasons To Reject:**

1. The experiment results cannot support the authors' argument well. The INT4 AQAS shows similar results as INT4 SQ in all experiments carried out in the paper. There is no good justification for using AQAS rather than SQ.

2. This paper lacks of novelty. I understand that this paper focuses on LLMs, but there are similar investigations around W4A8 PTQ for language models: https://openreview.net/pdf?id=tvDRmAxGIjw, and clamping denormal strategy: https://arxiv.org/pdf/1909.13271.pdf.

**Reproducibility:**

3: Could reproduce the results with some difficulty. The settings of parameters are underspecified or subjectively determined; the training/evaluation data are not widely available.

**Reviewer Confidence:**

3: Pretty sure, but there's a chance I missed something. Although I have a good feel for this area in general, I did not carefully check the paper's details, e.g., the math, experimental design, or novelty.

---

> ### Author Rebuttal · Authors · 2023-08-29
>
> We sincerely appreciate the insightful feedback provided by the reviewer. In response to the raised questions, our answers are as follows.
>
> > **Reasons To Reject 1.** The experiment results cannot support the authors' argument well. The INT4 AQAS shows similar results as INT4 SQ in all experiments carried out in the paper. There is no good justification for using AQAS rather than SQ.
>
> Thank you for letting us clarify the results of AQAS compared to SQ. We claim that AQAS performs robustly across various LLMs with distinct characteristics (OPT vs LLaMA) for W4A8 quantization, while SQ barely manages to retain comparable accuracy for OPT (only with the help of OPTQ), it still suffers noticeable accuracy degradation on LLaMA and LLaMA-2.
>
> To elucidate the superior performance of AQAS over SQ, consider the following points:
>
> - As shown in Table 3, SQ alone falls short for W4A8, exhibiting PPL scores of 1596.83 and 897.25 on OPT-6.7B/13B, respectively.
> - While OPTQ assists SQ in achieving accuracy comparable to AQAS on OPT, SQ still suffers a noticeable performance drop on LLaMA-7B. Specifically, SQ shows a 2.58 PPL degradation compared to AQAS in wikitext as shown in Table 3.
> - AQAS's balanced scaling strategy ensures robust accuracy for both OPT and LLaMA. For instance, it exhibits PPL scores of 12.19 and 12.08 with and without OPTQ on OPT-13B and demonstrates a 1.03 higher PPL than SQ on LLaMA. Similar trends are also observed across the CSQA and MMLU tasks in Tables 4 and 5.
> - We newly added experimental results of the recently released LLaMA-2 model, as observed in Table R1, AQAS outperforms SQ in all tasks, regardless of whether OPTQ is applied. Importantly, the significant accuracy degradation of SQ on MMLU and CSQA tasks seen with LLaMA is consistently evident with LLaMA-2 as well.
>
> The observed unstable performance of SQ on LLaMA models is because of the instability of SQ, **which unilaterally shifts the quantization difficulty by moving activation outliers to the weights**, showing varying performance depending on the models and the assistance of OPTQ. In contrast, AQAS, which explores **scaling while considering the quantization results of both activation and weight**, delivers more consistent and reliable performance across different models, underscoring its superiority in the W4A8 quantization.
>
> LLaMA-2-7B (W4A8) | WikiText-2 | MMLU | CSQA
> -- | -- | -- | --
> Weight Scaling | PPL | Acc | Acc
> FP | 5.42 | 41.6 | 71.76
> SQ | 6.82 | 38.6 | 68.55
> AQAS | 6.46 | 40.7 | 70.40
> SQ+OPTQ | 6.65 | 38.9 | 69.51
> AQAS+OPTQ | **6.45** | **41.1** | **70.94**
>
> Table R1. Performance comparison between SQ and AQAS across wikitext, MMLU, and CSQA (PIQA, winogrande, ARC_easy) tasks with LLaMA-2-7B.
>
> To gain a deeper understanding of the differences between SQ and AQAS, we conducted an analysis focusing on the scale values and their corresponding layer output quantization errors (via SQNR)  in both methods, as presented in Table R2. Notably, in Layers 5 and 10, AQAS employs smaller scale values than SQ. These **contrasting scale values lead to substantially higher output quantization errors** in the initial layers for SQ compared to AQAS. Even though similar scale values emerge in later layers, these variations result in markedly different text generation performance, as evidenced by the PPL comparisons between SQ and AQAS in Table R2.
>
> OPT-6.7B | Layer | 5 | 10 | 20 | 30 | W4A8 PPL
> -- | -- | -- | -- | -- | -- | --
> SQ | Scale | 22.55 | 22.39 | 25.92 | 31.53 | 1586.83
> SQ | SQNR | 12.51 | 4.29 | 11.47 | 9.71 | -
> AQAS | Scale | 13.44 | 14.91 | 21.69 | 28.45 | **13.42**
> AQAS | SQNR | **16.76** | **5.91** | **11.90** | **9.93** | -
>
> Table R2. Comparison of scale values for outlier channel and per-layer output SQNR between SQ and AQAS in OPT-6.7B and wikitext-2 PPL results of W4A8 PTQ OPT-6.7B
> (SQNR: signal-to-quantization-noise ratio, higher the better)
>
> &nbsp;
>
> > **Reasons To Reject 2.** This paper lacks of novelty. I understand that this paper focuses on LLMs, but there are similar investigations around W4A8 PTQ for language models: https://openreview.net/pdf?id=tvDRmAxGIjw, and clamping denormal strategy: https://arxiv.org/pdf/1909.13271.pdf.
> **Questions.** What is the unique contribution of this paper compared with similar investigations around W4A8 PTQ for language models: https://openreview.net/pdf?id=tvDRmAxGIjw, and clamping denormal strategy: https://arxiv.org/pdf/1909.13271.pdf?
>
> We thank the reviewer for letting us clarify the novelty of our work over the suggested related works.
>
> We would like to emphasize that our work proposes the **first comprehensive study to tackle unprecedented W4A8 quantization for large (Transformer-decoder-based) language models with tens of billions of parameters**. To overcome unique quantization challenges of systematic outliers (Dettmers et al., 2022) in LLMs, we comprehensively improve three key aspects of quantization, 1) scaling for quantization dynamic ranges, 2)  task-specific sequence-length calibration, and 3) number representation format for avoiding underflow, thereby extending the boundaries of weight and activation quantization of LLMs. Specifically, our contributions are as follows:
>
> - Analyzing the characteristics of activations and weights across different LLMs to address the challenges of activation/weight PTQ, thereby proposing a new quantization scaling method (AQAS)
> - Proposing a target task adaptive calibration methodology to offer a task-specific PTQ approach (SLAC).
> - Introducing the dINT format aimed at solving the previously overlooked issue of underflow in low-precision computations (dINT).
>
> Here's how each of these contributions contrasts with the papers you suggested:
>
> **[Comparison with W4A8 PTQ Paper [R1] https://openreview.net/pdf?id=tvDRmAxGIjw ]**
>
> - Unlike [R1], which focuses on encoder-only models, we perform a deep-dive into the challenges of low-precision PTQ for decoder-only LLMs, which exhibits unique characteristics of systematic outliers (Dettmers et al., 2022).
> - Our scaling methods AQAS, designed to be computationally viable for billion-scale LLMs, contrast with [R1]'s back-propagation-dependent approach, which is revealed to be unsuitable (Frantar et al., 2023).
> - Diverging from [R1]'s task-agnostic methods, we introduce a task-specific PTQ calibration technique called SLAC, which we showed critical for supporting duverse tasks.
>
> **[Comparison with AdaptivFloat [R2] https://arxiv.org/pdf/1909.13271.pdf ]**
>
> - [R2] did not investigate the trade-off between underflow and rounding errors in terms of quantization error, nor did it explore the necessity of denormal. As revealed in our study, this trade-off constitutes a critical determinant, especially in contexts of low precision, such as 4-bit precision.
> - To highlight the novelty of our approach, we conducted a brief ablation study on various formats of FP4 to demonstrate the significance of exploring the trade-off between underflow and rounding errors.
>    - Table R3 shows the breakdown of the mean-squared error (MSE) at the output of input-weight multiplication (cf. Eq. 4). While FP4 1-2-1 has the least rounding error, it exhibits significant underflow error. Conversely, FP4 1-3-0 reduces underflow error but has elevated rounding error.
>    - dINT strikes a balance between these errors, resulting in enhanced inference accuracy across wikitext, PIQA, and MMLU tasks as shown in Table R4.
>
> | W4 Format | Total error | by Rounding error | by Underflow error | by Cross-term error
> |-- | -- | -- | -- | --
> |FP4 (1-2-1) | 39.4017 | **4.2784** | 34.6510 | 0.4723
> |FP4 (1-3-0) | 17.6443 | 17.6315 | **0.0163** | **0.0035**
> |dINT4 | **7.9299** | 7.4314 | 0.4594 | 0.0390
>
> Table R3. Analysis of output quantization error (MSE) in W4A16 with LLaMA-7B single layer with breakdown by weight quantization error using Eq.4. (Layer 15 Query projection, unit: 1e-3)
>
> Model | Precision | W4 Format | Wikitext (PPL ↓) | PIQA (ACC ↑) | MMLU (avg ↑)
> -- | -- | -- | -- | -- | --
> LLaMA-7B | FP16 | - | 5.68 | 78.29 | 35.20
> |  | W4A16 | FP4 (1-2-1) | 26.52 | 62.84 | 27.31
> |  |   | FP4 (1-3-0) | 6.30 | 76.77 | 31.46
> |  |   | dINT4 | **6.07** | **77.91** | **32.53**
>
> Table R4. Experimental results of weight-only quantization (W4A16) utilizing FP4 variants and dINT4
>
> We are grateful for the reviewer’s feedback which has significantly enhanced the clarity of the comparison between related work and our contributions.
>
> [R1] Bai et al, “Towards Efficient Post-training Quantization of Pre-trained Language Models” NeurIPS 2022
> [R2] Tambe et al, “Algorithm-Hardware Co-Design of Adaptive Floating-Point Encodings for Resilient Deep Learning Inference” DAC 2020

---

### Official Review · Reviewer_WZ3Z · 2023-08-09

**Soundness:** 4

**Excitement:**

4: Strong: This paper deepens the understanding of some phenomenon or lowers the barriers to an existing research direction.

**Paper Topic And Main Contributions:**

This work focuses on LLMs' post-training quantization (PTQ), especially W4A8 quantization, to improve the efficiency of LLMs. The two core techniques proposed in this work are activation-quantization-aware scaling (AQAS) and sequence-length-aware calibration (SLAC). In addition, a novel data format called dINT is introduced to solve the underflow challenges effectively. The method is evaluated across different LLMs (OPT, LLaMas) and various tasks and metrics (PPL on language modeling, zero-shot reasoning tasks, few-shot MMLU, etc.) to demonstrate effectiveness and efficiency.

**Questions For The Authors:**

Please see the *Reasons To Reject section.

**Reasons To Accept:**

1. The paper is well-written and easy-to-follow. The visualization of the activation range (Fig.1) and quantization error (Fig.4) are helpful in understanding the advantage of the proposed method and data format.
2. Insights on the dynamic range of different layers and variations with different sequence lengths are provided. These findings are helpful for future research on LLM quantization (both PTQ and QAT).
3. Comprehensive experimental design across various LLMs and tasks. These results clearly demonstrate the effectiveness of the proposed method.
4. The evaluation of the hardware performance of MAC units on dINT4 and INT8 is informative.

**Reasons To Reject:**

1. The method is only verified on the W4A8E4 setting. However, previous work (SmoothQuant, AWQ) can be applied with various precision.
2. Some comparisons may be unfair:
    - In Table 2, comparing the area and power of INT8-based MAC units and dINT4-based MAC units is unfair.
    - In Figure 5 the FP4 is shown with mantissa=1, exponent=3, and no bias bit (correct me if I am wrong), which is not an optimal format for FP4. Also, I guess the error will be lower than dINT4 if the FP4 quantization error is visualized in Figure 4.

**Reproducibility:**

4: Could mostly reproduce the results, but there may be some variation because of sample variance or minor variations in their interpretation of the protocol or method.

**Reviewer Confidence:**

4: Quite sure. I tried to check the important points carefully. It's unlikely, though conceivable, that I missed something that should affect my ratings.

---

> ### Author Rebuttal · Authors · 2023-08-29
>
> We appreciate the reviewer’s valuable feedback. Responses to your comments are provided below.
>
> > **Reasons To Reject 1.** The method is only verified on the W4A8E4 setting. However, in previous work SmoothQuant(SQ), AWQ can be applied with various precision.
>
> Thank you for the valuable comment on the necessity of experiments across various precisions. Following the reviewer’s advice, we extended our experiments to include a range of precisions (W8A8, W3A16, and W3A8) studied in the previous work (SQ, AWQ). As shown in Table R1, the proposed AQAS and dINT achieve consistent accuracy gain for aggressive 3-bit weight AND activation quantization.
>
> | Wikitext-2 | OPT/LLaMA | W8A8 | W8A8 | W3A16 | W3A16 | W3A8 | W3A8
> -- | -- | -- | -- | -- | -- | -- | --
> Scaling method | Weight format | OPT-6.7B | LLaMa-13B | OPT-6.7B | LLaMa-13B | OPT-6.7B | LLaMa-13B
> | - | FP Baseline | 12.29 | 5.09 | 12.29 | 5.09 | 12.29 | 5.09
> SQ | INT | 12.31 | 5.13 | 50053.79 | 142.54 | 48777.33 | 123.71
> AWQ | INT | 99.40 | 5.15 | 17.60 | 7.43 | 3013.52 | 7.53
> AQAS | INT | **12.29** | **5.11** | 16.63 | 6.34 | 17.96 | 6.47
> AQAS | dINT | **12.29** | **5.11** | **14.39** | **5.99** | **15.10**| **6.08**
>
> Table R1. Comparison of SQ, AWQ, and AQAS scaling methods for various PTQ precisions (Wikitext 2 PPL, the lower the better).
>
> - In the case of W8A8, AWQ's lack of consideration for activation quantization results in significant performance degradation, particularly in models with notable activation outliers like OPT. Although SQ outperforms AWQ by scaling activation outliers, it still falls short of AQAS, which delivers performance nearly on par with full-precision models.
>
> - In the case of W3A16 (weight-only quantization), AWQ specialized for low-precision weight-only quantization significantly recovers accuracy compared to SQ, which suffers from a severe accuracy drop. However, AWQ still exhibits noticeable accuracy loss at 3-bit weight quantization. Conversely, AQAS achieves performance that is much closer to full-precision models. Note that the dINT format significantly improves weight quantization performance in sub-4-bit precision scenarios.
>
> - In more aggressive bit precision settings for both weight AND activation quantization (i.e., W3A8), both SQ and AWQ exhibit a substantial accuracy drop in both OPT and LLaMA. Conversely, AQAS and dINT effectively mitigate this accuracy degradation, particularly in the OPT model, bringing results nearly in line with full-precision baselines. This observation highlights the combined efficacy of AQAS in adjusting quantization scales and dINT in enhancing the representation of small values for aggressively precision-scaled PTQ.
>
> &nbsp;
>
> > **Reasons To Reject 2-1.** In Table 2, comparing the area and power of INT8-based MAC units and dINT4-based MAC units is unfair.
>
> Thank you for allowing us to elucidate the comparison between INT8-INT8 and dINT4xINT8 units in Table 2. Our study primarily explored whether the dINT4-based MAC unit could offer performance gains proportionate to the bit-precision reduction over the INT8-based unit. As noted in previous research [R1], lowering bit-precision can proportionally enhance both area and power efficiency, leading to increased throughput (e.g., transitioning from FP8-FP8 to INT4-INT4 boosts throughput 4x, as per [R2]). Table 2 shows that by reducing weight precision from 8-bit to 4-bit, our dINT format yields 2x area and 2.5x power efficiency improvements, underscoring dINT's ability to offer scalable performance despite its representation (Eq. 5) distinct from INT.
>
> [R1] M. Horowitz. "Energy table for 45nm process", Stanford VLSI wiki. [Online]. Available: https://sites.google.com/site/seecproject
> [R2] Agrawal et al., “9.1 A 7nm 4-Core AI Chip with 25.6TFLOPS Hybrid FP8 Training, 102.4TOPS INT4 Inference and Workload-Aware Throttling”, ISSCC 2021
>
> &nbsp;
>
> > **Reasons To Reject 2-2.** In Figure 5 the FP4 is shown with mantissa=1, exponent=3, and no bias bit (correct me if I am wrong), which is not an optimal format for FP4. Also, I guess the error will be lower than dINT4 if the FP4 quantization error is visualized in Figure 4.
>
> Following the reviewer's suggestion, we conducted an ablation study on various FP4 bit configurations (sign-exponent-mantissa: 1-3-0, 1-2-1, 1-1-2). Note that instead of utilizing a distinct exponent bias, we have employed a scale factor similar to integer quantization for representing the maximum data magnitude. Benchmark results (Table R2) and quantization error analysis (Table R3) demonstrate that our chosen FP4 1-3-0 outperforms the other FP4 configurations, yet dINT further excels over FP4 1-3-0.
>
> Model | Precision | W4 format | Wikitext (PPL ↓) | PIQA (ACC ↑) | MMLU (avg ↑)
> -- | -- | -- | -- | -- | --
> |LLaMA-7B |   | FP16 baseline | 5.68 | 78.29 | 35.20
> |  | W4A16 | FP4 (1-1-2) | 165582.55 | 51.69 | 26.88
> |  |   | FP4 (1-2-1) | 26.52 | 62.84 | 27.31
> |  |   | FP4 (1-3-0) | 6.30 | 76.77 | 31.46
> |  |   | dINT4 | **6.07** | **77.91** | **32.53**
> LLaMA-13B |   | FP16 baseline | 5.09 | 78.78 | 47.15
> |  | W4A16 | FP4 (1-1-2) | 74763.98 | 52.29 | 24.72
> |  |   | FP4 (1-2-1) | 7.95 | 74.65 | 31.54
> |  |   | FP4 (1-3-0) | 5.56 | 78.62 | 40.76
> |  |   | dINT4 | **5.38** | **79.05** | **44.35**
> LLaMA-30B |   | FP16 baseline | 4.10 | 80.96 | 58.50
> |  | W4A16 | FP4 (1-1-2) | 34027.07 | 51.52 | 25.32
> |  |   | FP4 (1-2-1) | 9.10 | 71.22 | 32.05
> |  |   | FP4 (1-3-0) | 4.57 | 79.71 | 53.50
> |  |   | dINT4 | **4.36** | **80.41** | **55.87**
>
> Table R2. Experimental results of weight-only quantization (W4A16) utilizing FP4 variants and dINT4
>
> |  | Total error | by Rounding error | by Underflow error | by Cross-term error
> |-- | -- | -- | -- | --
> |FP4 (1-2-1) | 39.4017 | **4.2784** | 34.6510 | 0.4723
> |FP4 (1-3-0) | 17.6443 | 17.6315 | **0.0163** | **0.0035**
> |dINT4 | **7.9299** | 7.4314 | 0.4594 | 0.0390
>
> Table R3. Analysis of output quantization error (MSE) in LLaMa single layer with breakdown by weight quantization error using Eq.4. (LLaMa-7B Query projection Layer 15, unit: 1e-3)
>
> Here are key insights yielded from this ablation study:
>
> **Performance in benchmarks**: As evident from Table R2, employing a 1-bit exponent severely compromises performance, and even in the case of FP4 1-2-1, accuracy loss persists. While FP4 1-3-0 demonstrates robust performance compared to other FP4 options, our proposed dINT4 exhibits the best accuracy across all tasks.
>
> **Quantization error analysis**: Table R3 shows the breakdown of the mean-squared error (MSE) at the output of input-weight multiplication (cf. Eq. 4). While FP4 1-2-1 has the least rounding error, it exhibits significant underflow error. Conversely, FP4 1-3-0 reduces underflow error but has elevated rounding error. dINT strikes a balance between these errors, resulting in enhanced inference accuracy.

---

### Meta-Review · Area_Chair_VhVr · 2023-09-18

**Recommendation:** 4

**Metareview:**

This paper proposes a method to optimize the PTQ of LLMs. Initially, reviewers had several concerns regarding the clarity, less novelty, and unclear results comparison. But the authors have addressed them, therefore, I recommend acceptance of this paper.

---

### Decision · Program_Chairs · 2023-10-07

**Decision:**

Accept-Main

**Comment:**

This paper proposes a method to optimize the PTQ of LLMs. Initially, reviewers had several concerns regarding the clarity, less novelty, and unclear results comparison. But the authors have addressed them, therefore, I recommend acceptance of this paper.